# The Mediating Role of Cumulative Fatigue on the Association between Occupational Stress and Depressive Symptoms: A Cross-Sectional Study among 1327 Chinese Primary Healthcare Professionals

**DOI:** 10.3390/ijerph192315477

**Published:** 2022-11-22

**Authors:** Yushi Lu, Zhi Li, Yuting Fan, Jin Wang, Tian Zhong, Ling Wang, Ying Xiao, Dongmei Zhang, Qingsong Chen, Xi Yu

**Affiliations:** 1Faculty of Medicine, Macau University of Science and Technology, Avenida Wai Long Taipa, Macau 999078, China; 2School of Public Health, Guangdong Pharmaceutical University, Guangzhou 510310, China; 3National Institute of Occupational Health and Poison Control, Chinese Center for Disease Control and Prevention, Beijing 100050, China; 4Guangdong-Hong Kong-Macau Joint Laboratory for Contaminants Exposure and Health, Guangzhou 510006, China

**Keywords:** occupational stress, depressive symptom, cumulative fatigue, mediating effect, primary healthcare

## Abstract

Occupational stress and depressive symptoms are common among professionals in the primary healthcare system, and the former can lead to a more severe level of the latter. However, there are few studies on the mediating effect of occupational stress on depressive symptoms using cumulative fatigue as a mediating variable. The Core Occupational Stress Scale, the Self Diagnosis Scale of Workers’ Cumulative Fatigue, and the Patient Health Questionnaire were used in the proposed study. To analyze and test the mediating effect, the hierarchical regression analysis method and the Bootstrap method were applied. Our results showed that occupational stress was positively correlated with the level of cumulative fatigue (*p* < 0.01) and depressive symptoms (*p* < 0.01). Cumulative fatigue played a partial, mediating role between the four dimensions of occupational stress and depressive symptoms, and the effect size of occupational stress and each dimension was 0.116 (95% CI: 0.096–0.135, *p* < 0.001), −0.204 (95% CI: −0.245–−0.166, *p* < 0.001), 0.179 (95% CI: 0.143–0.218, *p* < 0.001), 0.333 (95% CI: 0.283–0.385, *p* < 0.001), and −0.210 (95% CI: −0.292–−0.132, *p* < 0.001), respectively, while the percentages of the mediating effects were 43.56%, 44.46%, 48.58%, 71.26%, and 45.80%, respectively. Occupational stress can directly or indirectly affect depressive symptoms through the mediating effect of cumulative fatigue. Therefore, primary healthcare professionals can reduce occupational stress, which in turn relieves depressive symptoms, and thus reduce cumulative fatigue levels.

## 1. Introduction

Occupational stress, also known as work pressure, is the psychological or physiological reaction that occurs when an individual’s position’s requirements cannot be met by his or her ability or resources, or the position does not match the individual’s needs. Previous studies have shown that long-term occupational stress damages the physical and mental health of individuals [1], such as the cardiovascular system and immune system, and lead to changes in mental state. According to past studies, the prevalence of stress, anxiety, and depression in medical staff is very high and common, among which depressive symptoms are more widespread [2]. Another survey found that 64.7% of doctors display depressive symptoms, and 41.2% have occupational stress [3]. A researcher in Australia assessed depressive symptoms and occupational stress among 102 nurses, and the findings revealed that the prevalence rates of depressive symptoms and occupational stress were 32.4% and 41.2%, respectively, with depressive symptoms being higher than the general population in Australia. The prevalence of 4% indicates that such diseases are more prevalent among medical personnel [4].

Depression is a mood disorder characterized by persistent low spirits, disappointment, loss of interest, feelings of worthlessness, and decreased sleep and appetite [5]. According to one study, occupational stress has a greater impact on the depressive symptoms of occupational groups, and addressing the factors that contribute to occupational stress can reduce the risk of depressive symptoms [6]. More than 7.02 million people in China have been infected with COVID-19 as of 17 September 2022, and more than 25,000 people have died from COVID-19 [7]. Following the COVID-19 pandemic, primary healthcare systems in China have played a significant role in epidemic monitoring and virus screening as part of regular epidemic prevention and control [8]. During the study’s investigation period, China was in the process of normalizing epidemic prevention and control. The employees of grassroots health systems would have additional epidemic prevention and control work, such as epidemic prevention health education and routine virus prevention and control, disinfection work, routine virus screening work, etc., in addition to their regular work. Strict epidemic prevention and control measures have kept the domestic case growth rate stable, and epidemic prevention and control have achieved remarkable results. However, the policy of regular epidemic has also brought huge workloads and pressure to primary healthcare professionals, subsequently affecting their physical and mental health [9]. Increased workload, physical exhaustion, and infection risk can all have a significant impact on the physical and mental health of primary healthcare workers. As a result, they are predisposed to mental health issues such as fear, anxiety, depression, and insomnia.

According to previous studies, occupational stress linked to symptoms such as depression and cumulative fatigue has a mediating effect on the relationship between occupational stress and depressive symptoms and is a decisive factor for depressive symptoms [10]. Cumulative fatigue refers to fatigue that cannot be recovered instantly and effectively. Alleviating cumulative fatigue can improve occupational stress and depressive symptoms. Therefore, this study investigated how primary health system workers’ cumulative fatigue affects the association between occupational stress and depressive symptoms, and proposed measures to reduce workers’ occupational stress, fatigue accumulation, and depressive symptoms. Some researchers conducted cross-sectional studies to examine the relationship between occupational stress, fatigue, and depression in call center employees and discovered that fatigue was a significant factor influencing the degree of employee depression and that occupational stress was associated with fatigue or depression symptoms. Significant correlations were found, but no studies were conducted to determine whether fatigue as a mediator influences the relationship between occupational stress and depressive symptoms. In addition, in the previous studies, no one studied the accumulated fatigue effect of primary healthcare professionals. In this study, primary healthcare system professionals were used as the research object for the first time, which improved previous research inadequacies, fully explored the relationship between occupational stress, cumulative fatigue, and depressive symptoms, and investigated the role of cumulative fatigue as a mediating variable between occupational stress and depressive symptoms.

The expected goals of this study are, firstly, to investigate the occupational stress, cumulative fatigue, and depressive symptoms of primary healthcare professionals; secondly, to analyze the relationship among occupational stress, cumulative fatigue, and depressive symptoms; thirdly, to explore the mediating role of cumulative fatigue between occupational stress and depressive symptoms; and lastly, to provide a theoretical basis for formulating health strategies for primary healthcare professionals and preventing cumulative fatigue, occupational stress, and depressive symptoms.

## 2. Materials and Methods

### 2.1. Participants

A multistage stratified cluster sampling method was used in this cross-sectional study [11,12]. Guangdong Province’s prefecture-level cities are classified into three levels based on the gross domestic product (GDP) of each prefecture-level city in the current year, as published by the Guangdong Provincial Bureau of Statistics. GDP > USD 125 billion, USD 28–125 billion, and GDP < USD 28 billion were recorded as the 3 levels. Additionally, 4 primary healthcare institutions (including health bureaus, centers for disease control and prevention, community health services centers, and public hospitals) were randomly selected at each GDP level [13,14]. The participants were investigated via an Internet questionnaire sent by the WeChat APPs. Additionally, all the employees of 12 primary healthcare institutions in 10 cities were selected as research subjects (*n* = 1430). The inclusion criteria of the research objects were 18 years old or above and continuous employment in their current position for at least half a year. The data were collected from September to December 2021. A total of 1327 valid questionnaires were collected in this survey, with a response rate of 92.8%.

### 2.2. Basic Investigation

The subjects’ demographic and job-related characteristics, such as gender, age, education level, marital status, monthly income, post, shift, night shift, weekly working hours, etc., were collected using the questionnaires.

### 2.3. Measurement of Occupational Stress

The Core Occupational Stress Scale (COSS) was used in this investigation, which was developed by the Institute of Occupational Health and Poisoning Control of the Chinese Center for Disease Control and Prevention [15]. The scale consists of 17 items divided into 4 dimensions: “social support” (1–5 items), “organization and reward” (6–11 items), “demand and effort” (11–15 items), and “autonomy” (16–17 items). The Likert 5-point scale was used to score each item as completely disagree (1 point), disagree (2 points), basically agree (3 points), agree (4 points), and strongly agree (5 points). The items related to “social support” and “autonomy” were reversely scored, and then the scores of each dimension and the total score were calculated. A total score ≥50 indicated “occupational stress”, and the higher the total score was, the higher the degree of an individual’s occupational stress [16]. Through the reliability test, Cronbach’s α coefficients of the whole scale and all the dimensions were 0.681, 0.882, 0.754, 0.841, and 0.832, respectively. The exploratory factor analysis included 17 items. The results showed that the KMO value was 0.835, and the χ² value of the Bartlett spherical test was 9541.23 (*p* < 0.001), which was suitable for factor analysis. The contribution rate of the common factor cumulative variance of the total scale was 63.44%, and the factor load value of each item was in the range of 0.488–0.922. Each item and its factor load value are shown in Table 1.

### 2.4. Measurement of Depressive Systems

The Chinese translation of the Patient Health Questionnaire (PHQ-9) was used to evaluate the depressive symptoms of the subjects [17]. PHQ-9 is a self-assessment scale used to evaluate individuals’ depressive symptoms and depression degree. It consists of 9 items, and it is evaluated according to the frequency of symptoms in the past 2 weeks [18]. The Likert 4-point scale was used to score each item as never (0 points), occasionally (1 point), more than half (2 points), and always (3 points). A total score ≥10 meant “depressive symptoms”, and the higher the total score was, the higher the degree of an individual’s depressive symptoms. Through the reliability test, Cronbach’s α coefficient of the whole scale was 0.874. The exploratory factor analysis included 9 items. The results showed that the KMO value was 0.918, and the χ² value of the Bartlett spherical test was 4510.44 (*p* < 0.001), which was suitable for factor analysis. The contribution rate of the common factor cumulative variance of the total scale was 50.22%, and the factor load value of each item was 0.542–0.788, as shown in Table 2.

### 2.5. Measurement of Cumulative Fatigue

The Self-Diagnosis Scale of Workers’ Cumulative Fatigue, issued by the Ministry of Health, Labor, and Welfare of Japan, was used for this investigation [19]. This is the most widely used scale in China for measuring the overworked state of the occupational population. Some studies have used this scale to survey a machinery manufacturing enterprise to prevent employees from becoming overworked [20]. The scale includes two dimensions “self-conscious symptoms evaluation” and “work condition evaluation”. “Self-conscious symptoms evaluation” includes 13 items. After calculating the total score of each item, <5 was defined as Grade I; 5–10 was defined as Grade II; 11–20 was defined as Grade III; and ≥21 was defined as Grade IV. “Work condition evaluation” includes seven items. After calculating the total score of each item, 0 was defined as Grade A; 1–2 was defined as Grade B; 3–5 was defined as Grade C; and ≥6 was defined as Grade D. Finally, according to the “work burden score table”, combined with the grading of the two dimensions, the total score of cumulative fatigue was obtained. Additionally, the higher the total score, the greater the cumulative fatigue of the individuals. A total score ≥2 meant “cumulative fatigue”. Through the reliability test, Cronbach’s α coefficients of the whole scale and the two dimensions were 0.892, 0.895, and 0.711, respectively. The exploratory factor analysis included 20 items. The results showed that the KMO value was 0.921, and the χ² value of the Bartlett spherical test was 9981.76 (*p* < 0.001), which was suitable for factor analysis. The contribution rate of common factor cumulative variance of the total scale was 51.07%, and the factor load value of each item was in the range of 0.439–0.852. Each item of the scale and its factor load value are shown in Table 3.

### 2.6. Statistic Analysis

The information from the collected questionnaires was sorted out, and more than 20% of the missing items from the questionnaires were eliminated. Epi Data 3.1 was used to repeat the entry of the questionnaire data by two persons separately and create a database. SPSS 22.0 (IBM, Armonk, NY, USA) was used to statistically analyze the data. The score of each questionnaire did not meet the normal distribution; thus, we used median (Q1, Q3) for statistical description. Then, the Mann–Whitney U test was used to compare the scores of “gender”, “marital status”, “shift or not”, and “night shift”. Additionally, the Kruskal–Wallis H test was used to compare the scores of “age”, “monthly income”, “occupation”, and “weekly working hours”. The Spearman correlation was used to analyze the relativity among occupational stress, cumulative fatigue, and depressive symptoms. The model included depressive symptoms as dependent variables, basic characteristics, four dimensions of occupational stress (“social support”, “organization and report”, “demand and effort”, “autonomy”), and cumulative fatigue as independent variables, and the relationship between the variables was interpreted using hierarchical regression analysis. Finally, the mediating effect of cumulative fatigue between occupational stress and depressive symptoms was tested by the Bootstrap method, with a two-tailed test α = 0.05. In this study, the mediation effect framework was based on the theory of mediating effects [21], in which “c” was the total effect of the independent variable of the occupational stress on the dependent variable of the depressive symptoms; “a” was the effect of the independent variable of the occupational stress on the mediating variable of the cumulative fatigue; “b” was the effect of the mediating variable of the cumulative fatigue on the dependent variable of the depressive symptoms, and “c′” was the direct effect of the independent variable of the occupational stress on the dependent variable of the depressive symptoms after the introduction of the mediating variable of the cumulative fatigue [22]. As shown in Figure 1, the control variables of this study are “marital status”, “monthly income”, “weekly working hours”, and “occupation”. The solid lines represent the effects of the dependent variables, independent variables, and mediating variables, and the dotted lines represent the effects of the control variables.

## 3. Results

### 3.1. Basic Information of Participants

Among the study participants, 378 were males (28.5%), and 949 were females (71.5%). Additionally, 634 (47.1%) professionals had a junior college degree or below, while 693 (52.9%) professionals had a bachelor’s degree or above. The number of doctors, nurses, medical technicians, or others was 428 (31.4%), 548 (42.1%), and 351 (26.5%), respectively. Additionally, 582 (43.9%) and 550 (44.1%) professionals worked in shifts and had night shifts, respectively.

### 3.2. Factors of Occupational Stress, Cumulative Fatigue, and Depressive Symptoms

As shown in Table 4, the occupational stress score was 45.0 (40.0, 50.0), and the detection rate was 27.5%. The differences in occupational stress scores among monthly income, post, weekly working hours, shift, and night shift were statistically significant (*p* < 0.05). Among them, those who earned less than USD 700 per month, worked more than 49 h per week, and worked in shifts and had night shifts had higher occupational stress scores.

The cumulative fatigue score was 2.0 (0.0, 4.0), and the detection rate was 57.5%. There were statistically significant differences in the cumulative fatigue scores for different ages, educational levels, posts, weekly working hours, shifts, and night shifts (*p* < 0.05). Among them, those 31–40 years old, with a bachelor’s degree or above, working as doctors, working more than 49 h weekly, and working in shifts and having night shifts had higher cumulative fatigue scores.

The depressive symptoms scored 8.0 (5.0, 10.0), and the detection rate was 29.0%. There were statistically significant differences in depressive symptoms among different ages, marital status, monthly incomes, posts, weekly working hours, and night shifts (*p* < 0.05). among them, those less than 30 years old, unmarried, with a monthly income of less than USD 700, working as doctors, working more than 49 h weekly, and working during night shifts had higher scores of depression symptoms.

### 3.3. Correlation of Occupational Stress, Cumulative Fatigue, and Depressive Symptoms

The correlation analysis results showed that occupational stress and its “organization and reward” and “demand and effort” dimensions were positively correlated with cumulative fatigue (*r* = 0.546, 0.438, 0.526, *p* < 0.01). The dimensions of “social support” and “autonomy” were negatively correlated with cumulative fatigue (*r* = −0.394, −0.135, *p* < 0.01). Occupational stress and its dimensions of “organization and reward” and “demand and effort” were positively correlated with depressive symptoms (*r* = 0.514, 0.365, 0.362, *p* < 0.01). The “social support” and “autonomy” dimensions were negatively correlated with depressive symptoms (*r* = −0.394, −0.135, *p* < 0.01). The cumulative fatigue was positively correlated with depressive symptoms (*r* = 0.566, *p* < 0.01). See Table 5 for details.

### 3.4. Stratified Regression Analysis on Cumulative Fatigue, Occupational Stress, and Depressive Symptoms

The results of the multicollinearity analysis had a tolerance of 0.468–0.865 and VIF of 1.156–2.139, so there was no multicollinearity between the variables. Taking the depressive symptom score as the dependent variable, firstly, the *p* < 0.05 in the linear regression analysis was incorporated in the regression model, and the results showed that marital status, monthly income, weekly working hours, and the post had statistical significance on depressive symptoms (*p* < 0.05). Secondly, the four dimensions of occupational stress (“social support”, “organization and reward”, “demand and effort”, and “autonomy”) were included in the regression model as the second layer of variables, and the results showed that the scores of “organization and reward” and “demand and effort” were positively correlated with the depressive symptom scores (*β* = 0.182, 0.321, *p* < 0.05), and the scores of “social support” and “autonomy” were negatively correlated with the depressive symptom scores (*β* = −0.329, −0.265, *p* < 0.05), and the variability of depressive symptoms in the four dimensions of occupational stress was 24.3%. Thirdly, based on step 2, the cumulative fatigue scores were incorporated into the model, and the results showed that the cumulative fatigue scores were positively correlated with the depressive symptom scores (*β* = 0.889, *p* < 0.05), and cumulative fatigue was interpreted as 11.6% of the results of depressive symptoms, as shown in Table 6.

### 3.5. Mediation of Cumulative Fatigue between Occupational Stress and Depressive Symptoms

The Bootstrap method was used to test the mediating effect of cumulative fatigue between occupational stress and depression symptoms, and the judgment basis was whether 0 was included in the 95% confidence interval. The independent variables were occupational stress and its dimension scores, the dependent variable was depressive symptom scores, the mediating variable was cumulative fatigue scores, and the control variables were marital status, monthly income, posts, and weekly working hours. The total effects of occupational stress on depressive symptom scores were divided into direct effects and indirect effects (i.e., mediating effects). The total effect was the effect of the independent variable of occupational stress on the dependent variable of depressive symptoms when the mediating variable was not controlled. The direct effect was the effect of the independent variable of occupational stress on the dependent variable of depressive symptoms after controlling the mediating variable. The indirect effect was the effect of the independent variable of occupational stress on the depressive symptom scores of the dependent variable through the mediating variable cumulative fatigue scores. As shown in Table 4, 0 was not included in the 95% confidence interval of the mediating effect of the cumulative fatigue between occupational stress and its dimensions (“social support”, “organization and reward”, “demand and effort”, and “autonomy”) and depressive symptoms, which meant a mediating effect. The cumulative fatigue played a partial mediating role between occupational stress and depressive symptoms, and the mediating effect values in occupational stress and various dimensions were 0.116 (95% CI: 0.096–0.135, *p* < 0.001), −0.204 (95% CI: −0.245–−0.166, *p* < 0.001), 0.179 (95% CI: 0.143–0.218, *p* < 0.001), 0.333 (95% CI: 0.283–0.385, *p* < 0.001), and −0.210 (95% CI: −0.292–−0.132, *p* < 0.001); the percentage of mediating effects was 43.56%, 44.46%, 44.58%, 71.26%, and 45.80%, respectively, as detailed in Table 7. The mediating effect model is shown in Figure 2. Among them, the effect value of social support on cumulative fatigue was −0.197 (*p* < 0.05), and the direct effect value on depressive symptoms was −0.225 (*p* < 0.05); the effect value of organization and reward on cumulative fatigue was 0.169 (*p* < 0.05), and the direct effect value on depressive symptoms was 0.190 (*p* < 0.05); the effect value of demand and effort on cumulative fatigue was 0.309 (*p* < 0.05), and the direct effect value on depressive symptoms was 0.134 (*p* < 0.05); the effect value of autonomy on cumulative fatigue was −0.183 (*p* < 0.05), and the direct effect value on depressive symptoms was −0.249 (*p* < 0.05); the effect value of cumulative fatigue on depressive symptoms was 0.869 (*p* < 0.05).

## 4. Discussion

### 4.1. Effect of Occupational Stress on Depressive Symptoms

The results of this study showed that the detection rate of occupational stress among primary healthcare professionals was 27.5%, which was similar to the detection rate of occupational stress measured by Jin W. et al. [16], using the same COSS scale (27.0%) and similar to that (28.6%) measured by Marijanović I. et al. [23], with the DASS-21 scale, but higher than that measured by Soteriades E.S. et al. [24], with the DASS-S scale (11.0%). The results of this study showed that low monthly income, being a doctor, long weekly working hours, the need for shifts, and the need for night shifts were the key factors leading to occupational stress among primary healthcare professionals, which were consistent with the existing research results [25,26].

The detection rate of the depressive symptoms in the primary healthcare professionals in this study was 29.0%, which was similar to that of Lasalvia A. et al. [27] when compared with the relevant findings using the same assessment scale (26.6%). It was higher than that measured by Sahimi H.M.S. et al. [28] (11.1%). The results of that study showed that younger age, unmarried, low monthly income, doctors, long weekly working hours, and the need for night shifts were the main factors in the depressive symptoms of primary healthcare professionals, which was also consistent with the present research results [29,30].

The above results showed that the current mental health of these professionals was relatively poor, and occupational stress and depressive symptoms were more common, which were related to the work requirements and working environments of the primary healthcare systems. The functions of the primary healthcare systems are extensive, involving clinical and nursing services, health education guidance, and the management of case reporting. Therefore, the workload of primary healthcare professionals is heavy, thus subjecting them to occupational stress. In addition, the dedication and the sense of service of healthcare professionals often make them neglect their personal needs in the process of work. If their self-sacrifice is extreme, it will lead to a psychological imbalance among primary healthcare professionals and psychological problems such as occupational stress and depressive symptoms. Coupled with the COVID-19 pandemic in recent years, and the regular epidemic policy in China, the work tasks of primary healthcare professionals become more arduous, and thus these multiple pressures lead to increasing occupational stress [31]. The results of this study showed that occupational stress and depressive symptoms of primary healthcare professionals are positively correlated, which was consistent with the existing research results [22]. This result suggests that the higher the occupational stress of primary healthcare professionals is, the more severe depressive symptoms are, so reducing occupational stress in the primary healthcare systems can effectively improve depressive symptoms.

At the same time, among the four dimensions of occupational stress, the dimensions of “organization and reward” and “demand and effort” were positively correlated with depressive symptoms, while the dimensions of “social support” and “autonomy” were negatively correlated with depressive symptoms. This showed that the primary healthcare professionals made great efforts owing to the higher work requirements, while they received fewer rewards from work and social support, with a low degree of autonomy, which was liable to cause depression in the primary healthcare professionals and affect their physical and mental health. Meanwhile, the results of the stratified regression analysis in Table 6 showed that the four dimensions of occupational stress increased the variation in the results of depressive symptoms by 24.3%, indicating that occupational stress in primary healthcare professionals can affect depressive symptoms. Therefore, it is recommended that healthcare system managers lessen the occupational stress of primary healthcare professionals by increasing employee salaries, decreasing overtime hours, and rationally planning shifts and night shifts, thereby reducing depressive symptoms.

### 4.2. The Mediating Role of Cumulative Fatigue

The detection rate of cumulative fatigue in this study was 57.5%, higher than that measured by Zhan Y.X. et al. [32] with the FS-14 scale (35.06%). However, it was lower than that measured by Tian F. et al. [33], with the FS-14 scale (83.70%). The results of that study showed that the cumulative fatigue scores of primary healthcare professionals were positively correlated with the depressive symptom scores, which was consistent with the present research results [34]. The chronic cumulative fatigue of primary healthcare professionals leads to low spirits and mental malaise and, in turn, leads to depression. The results showed that the factors of the cumulative fatigue of primary healthcare professionals were younger age, having a bachelor’s degree or above, working as a doctor, longer weekly working hours, the need for shifts, and the need for night shifts, which were consistent with the present research results [35,36]. Cumulative fatigue, as a mediating variable in this study, played a partial mediating role between the occupational stress and depressive symptoms of professionals in primary healthcare systems, which was consistent with the findings of Lin T.C. et al. [21]. The effect of occupational stress on depressive symptoms is weakened by cumulative fatigue. Reducing professionals’ stress levels may alleviate cumulative fatigue, lowering the severity of depressive symptoms. Reducing cumulative fatigue has been shown to effectively treat the depression caused by occupational stress in primary healthcare professionals.

The analysis of the mediating effect results showed that cumulative fatigue had a mediating effect between the “social support”, “organization and reward”, “demand and effort”, and “autonomy” dimensions of occupational stress and depressive symptoms, and the percentages of these mediating effects were 44.46%, 44.58%, 71.26%, and 45.80%, respectively. Combined with the analysis of the four dimensions of occupational stress, when primary healthcare professionals can gain the support and help of leaders and colleagues in time and obtain a high sense of work achievement and award, occupational stress can be reduced, fatigue accumulation can be alleviated, and negative emotions and depressive symptoms can be reduced. By contrast, if social support and resources are insufficient, it will aggravate the fatigue and depressive symptoms of professionals. The percentage value of the mediating effect between the demand and effort dimension (71.26%) and depressive symptoms was higher among them. This demonstrates that when demands and contributions are unequal, the effect of relieving cumulative fatigue on the improvement of depressive symptoms is stronger, and cumulative fatigue is more significant. At the same time, studies show that an imbalance between demands and efforts is a significant risk factor for depression in medical personnel. Inadequate pay and benefits, limited advancement opportunities, job insecurity, and heavy workloads have all become common occupational occurrences among primary healthcare professionals. Especially when the working environment and income of primary healthcare professionals cannot be quickly improved, relieving their cumulative fatigue can effectively improve their depressive symptoms.

The managers of healthcare systems can help professionals improve their health by advising them on work–life balance, adjusting work and living conditions, increasing physical activity, and improving physical quality. At the same time, healthcare system administrators should organize regular physical examinations for employees, for the early detection of physical problems, which can effectively prevent illness and fatigue. To improve work efficiency while ensuring professional health, the human resources management department of the primary healthcare system can adjust the workload or work tasks based on the results of the physical examination and the wishes of professionals. In addition, the managers of healthcare systems should pay attention to the psychological problems of primary workers and provide professional psychological consulting services for workers. It is also possible to organize a professional psychological counseling team to carry out group counseling activities for employees. While providing psychological support to professionals experiencing depressive symptoms, it is also necessary to protect their privacy, attend to their needs, and provide logistical services. At the same time, night shifts and overtime work should be reasonably arranged, and the recreational and leisure activities of professionals should be regularly arranged to reduce stress, alleviate the cumulative fatigue of primary healthcare professionals and reduce their depressive symptoms. Primary healthcare professionals should also pay attention to maintaining a balance between work effort and their own needs, with adequate exercise and rest to relieve fatigue, cultivating hobbies to enjoy life and relax work nerves, and seeking medical treatment in time when there is a long-term low spirit to prevent depression.

## 5. Conclusions and Prospects

In summary, primary healthcare professionals can alleviate their cumulative fatigue levels by reducing occupational stress, which in turn relieves depressive symptoms. Therefore, it is recommended that the management personnel of the healthcare systems regularly carry out psychological science lectures, give timely psychological support and help, and appropriately organize relaxation and leisure activities to alleviate cumulative fatigue. At the same time, health management should be provided for primary healthcare professionals, and physical examinations should be scheduled on a regular basis to prevent diseases and cumulative fatigue. Professionals with occupational stress or depressive symptoms should receive psychological counseling to improve their mental health.

This study is the first to use primary healthcare professionals as the research object to explore the impact of occupational stress on depressive symptoms. In addition, a mediating effect model was used in this study, and it was discovered that cumulative fatigue, as a mediating variable, played a partial mediating role between occupational stress and depressive symptoms among primary healthcare professionals. This study also provides a theoretical foundation for developing health strategies for primary care employees, preventing the occurrence of cumulative fatigue and occupational stress, as well as the onset of depression.

However, this was a cross-sectional study that did not determine the causal relationship among the four dimensions of occupational stress, cumulative fatigue, and depressive symptoms. Additionally, only primary healthcare workers in Guangdong Province participated in the study. To better explore the effects of occupational stress on depressive symptoms, subsequent research is suggested to expand the scope and sample size and make the findings more convincing. A nationwide survey of occupational stress and depressive symptoms among primary healthcare professionals could also be investigated. Additionally, the mediating role of cumulative fatigue should be explored as well.

## Figures and Tables

**Figure 1 ijerph-19-15477-f001:**
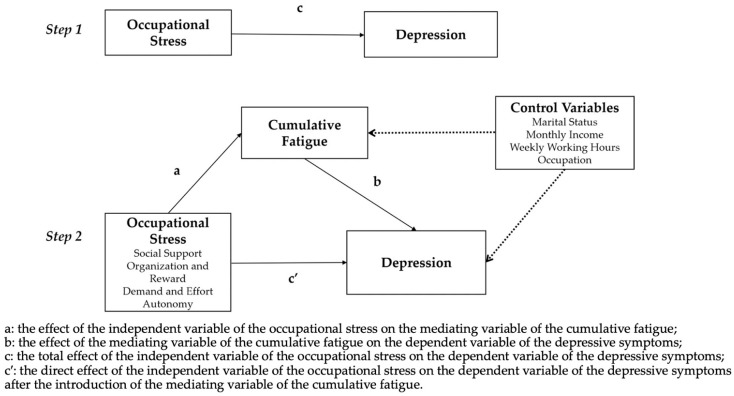
The mediating effect framework of cumulative fatigue between occupational stress and depressive symptoms.

**Figure 2 ijerph-19-15477-f002:**
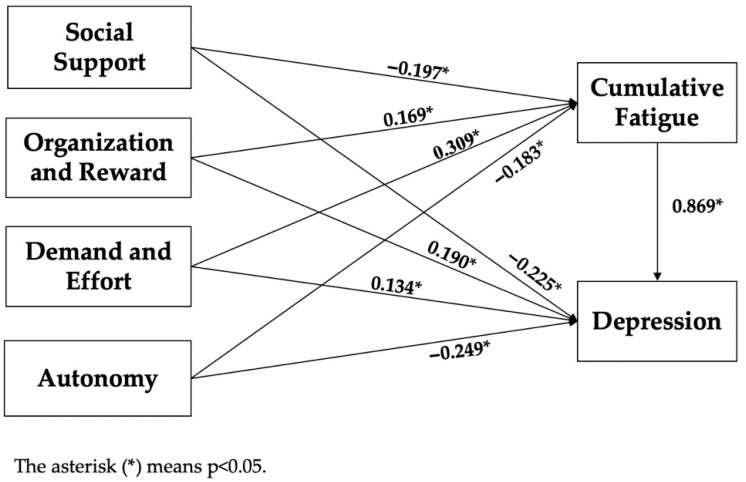
The mediating model of cumulative fatigue between occupational stress and depressive symptoms.

**Table 1 ijerph-19-15477-t001:** The Core Occupational Stress Scale with factor load values.

Number of Items	Factor 1	Factor 2	Factor 3	Factor 4
C7 Job–education commensurability	0.839			
C8 Effort–performance mismatch	0.821			
C9 Unnecessary changes to the job	0.789			
C6 Effort–respect mismatch	0.736			
C11 Hierarchical department	0.432			
C10 Cannot bear shift work				
C1 Get along well with leaders		0.853		
C2 Get along well with colleagues		0.838		
C3 The department is well-coordinated		0.819		
C4 Leader is helpful for work		0.817		
C5 Family supports my work		0.722		
C13 The job requires a quick pace			−0.834	
C15 Require to work extra hours			−0.829	
C12 Time is insufficient due to workload			−0.828	
C14 Job demands are getting higher			−0.773	
C16 Free to choose what to do at work				0.920
C17 Free to choose how to do at work				0.911

**Table 2 ijerph-19-15477-t002:** The Patient Health Questionnaire with factor load values.

Number of Items	Factor 1
D4 Feeling tired and bored	0.788
D2 Feeling bad, unhappy	0.764
D6 Feeling like a failure	0.745
D3 Can’t sleep or sleep too much	0.720
D8 Slowness or restlessness	0.713
D5 Poor appetite or eating too much	0.701
D1 Little or no interest in doing things	0.689
D7 Difficult in concentrating	0.688
D9 Thoughts of suicide or self-harm	0.542

**Table 3 ijerph-19-15477-t003:** The Self-Diagnosis Scale of Workers’ Cumulative Fatigue with factor load values.

Number of Items	Factor 1	Factor 2	Factor 3
G2 Disturbed	0.852		
G3 Restless	0.837		
G4 Feeling depressed	0.728		
G1 Irritable	0.697		
G5 Insomnia	0.627		
G7 Inattention	0.422		
G6 Poor health	0.414		
G14 Overtime in the past month		0.740	
G15 Irregular working hours		0.729	
G20 Physical burden of work		0.703	
G17 Burden of night shift		0.679	
G19 Mental burden of work		0.665	
G16 Burden of business trips		0.470	
G18 Rest time and facilities		0.439	
G9 Sleepy at work			−0.809
G10 Not motivated			−0.768
G12 Feeling tired in the morning			−0.691
G11 Exhausted			−0.657
G8 Error-prone			−0.623
G13 More fatigue than before			−0.600

**Table 4 ijerph-19-15477-t004:** Characteristics and scores of occupational stress, depressive symptoms, and cumulative fatigue among participants.

Variables	Amount	Occupational Stress			Cumulative Fatigue			Depression		
	(Ratio/%)	Total/Score	Z/H	*p*	Total/Score	Z/H	*p*	Total/Score	Z/H	*p*
Gender			−0.365	0.715		−1.633	0.102		−1.483	0.138
Male	378 (28.5)	44.5 (39.0,50.0)			2.0 (0.0,4.0)			8.0 (6.0,10.3)		
Female	949 (71.5)	45.0 (40.0,50.0)			2.0 (0.0,4.0)			8.0 (5.0,10.0)		
Age/Year			4.301	0.116		19.806	<0.001		13.732	0.001
≤30	416 (31.3)	45.0 (39.0,50.0)			2.0 (0.0,4.0)			8.0 (6.0,11.0)		
31–40	484 (36.5)	45.0 (40.0,51.0)			2.0 (0.0,4.0)			8.0 (5.0,10.0)		
≥41	427 (33.2)	44.0 (39.0,49.0)			1.0 (0.0,4.0)			7.0 (5.0,10.0)		
Education level			−0.760	0.448		−3.721	<0.001		−1.250	0.211
Junior college or below	634 (47.1)	45.0 (40.0,50.0)			2.0 (0.0,4.0)			7.5 (5.0,10.0)		
Bachelor or above	693 (52.9)	44.0 (39.0,50.0)			2.0 (0.0,4.0)			8.0 (5.0,10.0)		
Marital status			−0.492	0.623		−1.142	0.149		−4.122	<0.001
Unmarried	329 (24.8)	45.0 (39.0,50.0)			2.0 (0.0,4.0)			8.0 (6.0,12.0)		
Married	998 (75.2)	45.0 (40.0,50.0)			2.0 (0.0,4.0)			8.0 (5.0,10.0)		
Monthly income/USD			17.406	<0.001		1.053	0.591		10.254	0.006
≤700	488 (36.8)	46.0 (41.0,51.0)			2.0 (0.0,4.0)			8.0 (6.0,11.0)		
701–999	412 (31.0)	45.0 (40.0,50.0)			2.0 (0.0,4.0)			7.0 (5.0,10.0)		
≥1000	427 (32.2)	43.0 (38.0,49.0)			2.0 (0.0,4.0)			8.0 (5.0,10.0)		
Occupation			10.574	0.005		20.079	<0.001		10.158	0.006
Doctor	428 (32.3)	46.0 (40.0,51.8)			2.0 (0.3,4.0)			8.0 (6.0,11.0)		
Nurse	548 (41.3)	45.0 (40.0,50.0)			2.0 (0.0,4.0)			8.0 (5.0,10.0)		
Medical technician	351 (26.4)	44.0 (39.0,49.0)			1.0 (0.0,4.0)			8.0 (5.0,10.0)		
Weekly working hours/h			57.928	<0.001		149.455	<0.001		29.919	<0.001
≤40	290 (21.9)	43.0 (39.0,49.0)			0.0 (0.0,2.0)			7.0 (5.0,9.0)		
41–48	595 (41.8)	44.0 (38.0,49.0)			2.0 (0.0,4.0)			8.0 (5.0,10.0)		
≥49	442 (36.3)	47.0 (42.0,52.0)			4.0 (2.0,5.0)			8.0 (6.0,11.3)		
Shift or not			−4.644	<0.001		−4.642	<0.001		−0.572	<0.001
No	745 (56.1)	44.0 (39.0,49.0)			2.0 (0.0,4.0)			8.0 (5.0,10.0)		
Yes	582 (43.9)	46.0 (40.0,51.0)			2.5 (0.0,4.0)			8.0 (5.0,10.0)		
Night shift			−4.606	<0.001		−9.112	<0.001		−3.572	<0.001
No	777 (58.6)	44.0 (39.0,49.0)			1.0 (0.0,4.0)			7.0 (5.0,10.0)		
Yes	550 (41.4)	46.0 (40.0,51.0)			3.0 (1.0,4.3)			8.0 (6.0,10.0)		

**Table 5 ijerph-19-15477-t005:** Correlation analysis of occupational stress, cumulative fatigue, and depressive symptoms.

Variables	COSS	Social Support	Organization and Reward	Demand and Effort	Autonomy	Cumulative Fatigue	Depression
COSS	1.000						
Social support	−0.661 **	1.000					
Organization and reward	0.788 **	−0.351 **	1.000				
Demand and effort	0.644 **	−0.132 **	0.382 **	1.000			
Autonomy	−0.296 **	0.204 **	−0.078 **	−0.020	1.000		
Cumulative fatigue	0.546 **	−0.339 **	0.348 **	0.526 **	−0.107 **	1.000	
Depression	0.514 **	−0.394 **	0.365 **	0.362 **	−0.135 **	0.566 **	1.000

PS: ** means *p* < 0.01, and COSS means the Core Occupational Stress Scale, the same as below.

**Table 6 ijerph-19-15477-t006:** Stratified regression analysis on cumulative fatigue, occupational stress, and depressive symptoms.

Variables	Block 1		Block 2		Block 3	
	*β*	VIF	*β*	VIF	*β*	VIF
Marital status (unmarried as reference)						
Married	−1.060 **	1.060	−1.319 **	1.076	−0.955 **	1.087
Monthly income/USD (≤700 as reference)						
701–999	−0.774 **	1.328	−0.489	1.346	−0.385	1.347
≥1000	−0.98 **	1.414	−0.296	1.489	−0.396	1.490
Occupation (doctor as reference)						
Nurse	−0.700 *	1.425	−0.491 *	1.440	−0.505 *	1.440
Medical technician	−0.954 **	1.428	−0.417	1.447	−0.295	1.449
Weekly working hours/h (≤40 as reference)						
41–48	0.340	1.720	0.231	1.749	−0.170	1.767
≥49	1.425 **	1.731	0.344	1.874	−0.597 *	1.963
Occupational stress						
Social support			−0.329 **	1.180	−0.209 **	1.257
Organization and reward			0.182 **	1.322	0.140 **	1.335
Demand and effort			0.321 **	1.331	0.088 *	1.614
Autonomy			−0.265 **	1.077	−0.185 **	1.085
Cumulative fatigue					0.889 **	1.663
*F*	9.651		112.953		257.540	
Adjusted *R²*	0.044		0.286		0.403	
Δ*R²*	0.049		0.243		0.116	

* *p* < 0.05, ** *p* < 0.01 (two-tailed).

**Table 7 ijerph-19-15477-t007:** Analysis of the mediating effect of cumulative fatigue between occupational stress and depressive symptoms.

X Variables	c	a	b	c′	ab (95% CI)	Mediation Effect Percentage
Social support	−0.460 ***	−0.197 ***	1.040 ***	−0.255 ***	−0.204(−0.245~−0.166)	44.46
Organization and reward	0.369 ***	0.169 ***	1.061 ***	0.190 ***	0.179(0.143~0.218)	48.58
Demand and effort	0.467 ***	0.309 ***	1.076 ***	0.134 ***	0.333(0.283~0.385)	71.26
Autonomy	−0.459 ***	−0.183 ***	1.148 ***	−0.249 ***	−0.210(−0.292~−0.132)	45.80
COSS	0.266 ***	0.133 ***	0.869 ***	0.150 ***	0.116(0.096~0.135)	43.56

*** *p* < 0.001 (two-tailed), c = total effect value of X on Y; a = effect value of X on M; b = effect value of M on Y; c′ = direct effect value of X on Y; ab = intermediary effect value of depression; mediation effect percentage = ab/c × 100%.

## Data Availability

The datasets presented in this article are available from the corresponding author upon reasonable request.

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
