# Peer review of "The Mediating Role of Cumulative Fatigue on the Association between Occupational Stress and Depressive Symptoms: A Cross-Sectional Study among 1327 Chinese Primary Healthcare Professionals"

_ijerph, 2022, doi:10.3390/ijerph192315477_

Round 1
Reviewer 1 Report
This is a tight and well-argued paper. Most of my comments relate to questions I have in the “materials and methods” section of the paper.
· Did all of the employees of the 13 primary health institutions receive the survey?
· How were the medical and health institutions selected for analysis or was the populations of organizations used
· How was the survey distributed? Via email to a list of employees? Via a web link?
· I assume that “recovery rate” means “response rate” – is this the case?
· I would like to see a table that contains the questions used by category/concept along with the factor loadings.
· I am not sure I understand what is meant by “input the questionnaire data in parallel.”
· What two groups was the Mann-Whitney U test comparing?
· This may be beyond the scope of the paper but it would be very interesting to see what a path model looks like with the employee characteristics as the endogenous variables moving through the exogenous variables (social support, organization and reward, demand and effort, autonomy), through cumulative fatigue to depression.
· On a smaller note, there is a need for some grammatical editing throughout the paper.
Author Response
Point 1: Did all of the employees of the 13 primary health institutions receive the survey?
Response 1: Thank you for your question. We have double-checked the database for details and it is confirmed actually 12 institutions (apologies for the mistake writing of 13), 4 from each GDP level (3 levels in total), were included in the study. And yes, all the employees of the 12 primary health institutions did receive the survey. This is corrected and further explained in the revised manuscript. (see lines 114-115).
Point 2: How were the medical and health institutions selected for analysis or were the populations of organizations used?
Response 2: Thank you for your question. The multi-stage stratified cluster sampling method was used in this cross-sectional study and relevant references were added to support our method. More details about the sampling method we used have been added to the main text for the readers to better understand. As for the populations, yes, we did use the total populations of organizations used (n=1430). And this information has been added to the main text as well to let the readers better understand our sampling procedures. (See lines 106-113).
Point 3: How was the survey distributed? Via email to a list of employees? Via a web link?
Response 3: Thank you for your question. The participants were investigated via an Internet questionnaire sent by the Wechat APPs. The survey method was further explained in the main text as well, (lines 113-114).
Point 4: I assume that “recovery rate” means “response rate” – is this the case?
Response 4: Yes, it meant “response rate”. We have replaced it by “response rate” as suggested. (line 119).
Point 5: I would like to see a table that contains the questions used by category/concept along with the factor loadings.
Response 5: Thank you for your suggestion. We have added 3 tables of each scale’s questions and factor load values. (See line 143, Table 1, line 160, table 2, and line 185, table 3).
Point 6: I am not sure I understand what is meant by “input the questionnaire data in parallel.”
Response 6: We have corrected it by “entry the questionnaire data by two persons separately”. (lines 189-190).
Point 7: What two groups was the Mann-Whitney U test comparing?
Response 7: “Gender”, “Marital Status”, “Shift or Not”, and “Night Shift” were tested by the Mann-Whitney U test, as shown in Table 4 (each characteristic with two variables). We have rephrased the sentence and corrected our expression to make it clear for the readers (See lines 192-195).
Point 8: This may be beyond the scope of the paper but it would be very interesting to see what a path model looks like with the employee characteristics as the endogenous variables moving through the exogenous variables (social support, organization, and reward, demand and effort, autonomy), through cumulative fatigue to depression.
Response 8: Thank you for your suggestion. We have added the control variables of employee characteristics into the mediating effect framework as suggested. And their impacts have been further explained in the main text according to the new figure as well (See lines 211-214 and Figure 1).
Point 9: On a smaller note, there is a need for some grammatical editing throughout the paper.
Response 9: Thank you for your suggestion. We have asked the professionals for grammatical editing. We really hope the following manuscript has been substantially improved. (Line 50, 78, 118, 122,221, 223, 226, etc.)

Reviewer 2 Report
If the research aims to highlight any relations, it would be useful that those relations would be first described as research hypotheses, and theoretical and empirical arguments would be provided for each of them.
Please provide information on sampling procedure.
Line 175: It is not clear the meaning of “M (P25, P75)”
Line 182-193: Please provide references for the method employed for mediation.
Line 210: it would be useful to also display the income in a more commonly used currency.
Tables and figures should be inserted just after they are referred into the text.
Line 205-2020: The distribution of the three scores should be analysed in different paragraphs.
Line 305-358: The paragraph is longer than one page, which make it difficult to follow. Please divide it into smaller one. This issue may be found in other parts of the paper.
The research implications and directions of future research should be further outlined.
Author Response
We would like to thank the reviewer for offering these very constructive advice to help us improve oir work.
Point 1: Please provide information on sampling procedure.
Response 1: Thank you for your suggestion. The multi-stage stratified cluster sampling method was used in this cross-sectional study and relevant references were added to support our method. More details about the sampling method we used have been added to the main text for the readers to better understand. This is corrected and further explained in the revised manuscript (see lines 106-115).
Point 2: Line 175: It is not clear the meaning of “M (P25, P75)”
Response 2: Thank you for your suggestion. We have changed “M (P25, P75)” to “M (Q1, Q3)” in order to describe the quartiles more precisely. M refers to the median, Q1 refers to the median from the minimum to the median, and Q3 refers to the median from the median to the maximum. “M(Q1, Q3)” is used to describe non-normal measurement data. Such information has been added to the main text as well. ( line 192).
Point 3: Line 182-193: Please provide references for the method employed for mediation.
Response 3: Thank you for your suggestion. We have added references in this part. Please see page 6 of the revised manuscript ( line 204 and line 211).
Point 4: Line 210: it would be useful to also display the income in a more commonly used currency.
Response 4: Thank you for your suggestion. We have displayed the income in USD. ( line 229, line 241, Table 4, Table 6).
Point 5: Tables and figures should be inserted just after they are referred into the text
Response 5: Thank you for your suggestion. We have inserted all the tables after they are referred into the text.
Point 6: Line 205-2020: The distribution of the three scores should be analyzed in different paragraphs.
Response 6: Thank you for your suggestion. We have analyzed the distribution of three scores in different paragraphs. Please see line 226, line 231, and line 237.
Point 7: Line 305-358: The paragraph is longer than one page, which make it difficult to follow. Please divide it into smaller ones. This issue may be found in other parts of the paper.
Response 7: Thank you for your suggestion. We have divided the Discussion section into smaller parts according to the logic of writing. Please see lines 331, line 340, line 348, line 367, line 382, line 400, and line 420.
Point 8: The research implications and directions of future research should be further outlined.
Response 8: Thank you for your suggestion. We have included the research implications and directions of future research in the Conclusions and Prospects part as suggested. Please see lines 456-459, 462-463, and 465-467.

Round 2
Reviewer 2 Report
-